# RankMatch: A Novel Approach to Semi-Supervised Label Distribution Learning Leveraging Inter-label Correlations

## Abstract

This paper introduces RankMatch, an innovative approach to Semi-Supervised Label Distribution Learning (SSLDL). Addressing the challenge of limited labeled data, RankMatch effectively leverages a minimal set of labeled examples along with a substantial volume of unlabeled data, significantly reducing the manual labeling requirements for label distribution learning. Specifically, RankMatch employs an ensemble learning-inspired averaging strategy to generate pseudo-label distributions from multiple weakly augmented images, enhancing prediction stability and model robustness. Additionally, RankMatch incorporates a novel pairwise relevance ranking (PRR) loss to capture complex inter-label correlations, ensuring alignment of the predicted label distributions with the ground truth. We establish a theoretical generalization bound for RankMatch, and through extensive experiments, demonstrate its superiority in performance against existing SSLDL methods. *The code is available in the supplementary materials.*

## 1 Introduction

Label Distribution Learning (LDL) (Geng, 2016) is machine learning paradigm developed to address the issue of label ambiguity. Unlike Multi-label Learning (MLL) (Zhang & Zhou, 2014), LDL does more than assign a specific number of labels to each instance; it also quantifies the importance of each label. This additional metric, referred to as the label description degree (Geng, 2016; Jia et al., 2023), provides deeper semantic information, enhancing the interpretative richness of the data. For example, as demonstrated in Fig. 1,

an instance from a facial emotion dataset (Shih et al., 2008) is annotated not just with labels but with a distribution that specifies the relative importance of each emotion. This approach to labeling offers a more nuanced representation of real-world data. Recent advancements in LDL have significantly improved its application across various domains, such as expression recognition (Chen et al., 2020), facial age estimation (Geng et al., 2013), image object detection (Xu et al., 2023), joint acne image grading (Wu et al., 2019), and head-pose estimation (Liu et al., 2019). These developments underscore LDL's utility and effectiveness in practical settings.

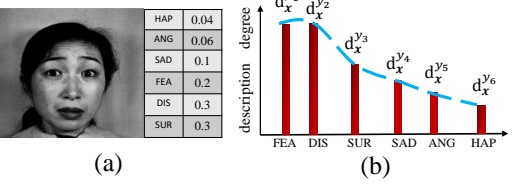

Figure 1: An illustration of an example from a facial SJAFFE dataset (Shih et al., 2008) annotated with a label distribution.

The success of deep learning heavily rely on large-scale and accurately labeled datasets, which are necessary to train very deep neural networks (DNNs) with superior generalization. However, acquiring such labeled data can be an arduous and costly process. Especially, it is more costly to obtain large dataset annotated with label distribution. For instance, considering the RAF-LDL dataset (Li & Deng, 2019), 315 trained annotators were employed, and each image is annotated for enough independent times to get the appropriate label distribution. As a result, the conflict emerges prominently when LDL embraces DNNs. A possible way to address the challenge is to leverage the highly available unlabeled data. In this paper, we attempt to address this issue by fundamentally

developing an LDL model that utilizes a small amount of labeled data along with a larger pool of unlabeled data.

Notice that semi-supervised learning (SSL) has already make significant advancements (Basak & Yin, 2023; Fini et al., 2023), especially in the era of deep learning. However, SSLDL has not been explored to the same extent. Traditional SSL approaches are mainly designed for single label learning or multi-label learning, which often rely on confidence-based pseudo-labeling (Jiang et al., 2022), (Sohn et al., 2020) and fall in Semi-Supervised Label Distribution Learning (SSLDL) because it aims to predict the whole label distribution, not just the most likely label. Moreover, exiting SSL methods typically ignore the correlation between labels (Xu & Zhou, 2017), potentially hindering their performance for LDL.

To address the complexities of SSLDL, this paper introduces a novel methodology termed RankMatch. This approach leverages an ensemble learning-derived averaging strategy (Zhou & Zhou, 2021) to compute the mean of predictions from variously augmented images (Sohn et al., 2020), thereby forming a robust pseudo-label distribution. Furthermore, to capture the correlations between labels, we introduce a new loss function called the pairwise relevance ranking loss (PRR loss). We apply a stringent version of PRR loss to labeled samples to ensure precise alignment with the ground-truth label distributions, and a version based on pseudo-label distributions for unlabeled samples. Essentially, both are designed to preserve the inherent label correlations on a rational basis, ensuring that the predicted distributions align with either ground-truth or pseudo-label distributions. In the theoretical analysis, we establish a generalization bound for RankMatch. Finally, in the experiments, we demonstrate that RankMatch can effectively address the SSLDL problem and outperform existing methods. In summary, our contributions can be summarized as

- We propose RankMatch, a novel approach that introduces a pairwise relevance ranking loss function, which captures inter-label correlations, effectively tackling the SSLDL problem.

- We provide a theoretical generalization bound for RankMatch, contributing to the understanding of SSLDL methods by analyzing their generalization capabilities.

- Through comprehensive experiments across multiple real-world datasets, we demonstrate that RankMatch consistently outperforms existing SSLDL methods.

## 2 RELATED WORK

### 2.1 LABEL DISTRIBUTION LEARNING

Label Distribution Learning (LDL) (Geng, 2016) assigns a range of labels to each instance, enabling a direct relationship between instances and their label distributions. Originally developed for facial age estimation (Geng et al., 2013), LDL generates distributions for all age categories, offering advantages over single-label approaches. This method is particularly effective in applications like facial emotion recognition, where it accurately represents complex emotional states by modeling the uncertainties within the label space (Xu & Zhou, 2017).

LDL's versatility extends to various applications. NASA, for instance, has used LDL to determine the chemical compositions of Martian meteorites (Morrison et al., 2018), fine-tuning the algorithm to predict elemental abundances from crystallographic data. In mental health, LDL has improved depression diagnosis through the Deep Joint Label Distribution and Metric Learning (DJ-LDML) method, which detects subtle facial expression variations across different depression levels (Zhou et al., 2020). Additionally, LDL has proven effective in static environments like indoor venues, where Ling (Ling & Geng, 2019) implemented it for crowd counting by assigning label distributions that accurately describe the crowd density in video frames.

### 2.2 SEMI-SUPERVISED LABEL DISTRIBUTION LEARNING

Lack of sufficient training data with exact labels is still a challenge for label distribution learning. To address this issue, several Semi-Supervised Label Distribution Learning (SSLDL) algorithms have been developed. For example, Hou (Hou et al., 2017) leverages the average labels from the neighbors of unlabeled data to determine its label distribution, then uses both labeled and unlabeled data to train the LDL model. Jia (Jia et al., 2021b) enhances label distribution recovery by harnessing

relationships among graph nodes. Liu (Liu et al., 2022) introduced a co-regularization based SSLDL algorithm that employs dual model structures to manage both labeled and unlabeled data, showing improved robustness and consistency.

While these SSLDL methods are varied, they generally do not provide an end-to-end solution. Traditional techniques often require manual intervention for feature engineering and struggle to handle large-scale, high-dimensional data effectively. They also fail to fully utilize unlabeled data. In contrast, deep learning excels in automatically learning complex features and has shown effectiveness in various data-rich environments. Therefore, there is significant interest in applying deep learning to overcome the inherent limitations of existing semi-supervised approaches and enhance the capabilities of SSLDL.

## 3 THE METHOD

### 3.1 PROBLEM STATEMENT AND NOTATION

In Semi-Supervised Label Distribution Learning (SSLDL), our training set, denoted by $\mathcal{D}$, comprises both labeled and unlabeled datasets: $\mathcal{D}_L = \{(\mathbf{x}_i, \mathbf{d}_i) | i \leq n\}$ contains labeled samples, and $\mathcal{D}_U = \{\mathbf{x}_g | g \leq m\}$ consists of unlabeled samples. Here, $\mathbf{x}$ represents an instance with the instance denoted by $\mathbf{x}_i$, and $\mathbf{d_i} = \{d_{\mathbf{x}_i}^{y_1}, d_{\mathbf{x}_i}^{y_2}, ..., d_{\mathbf{x}_i}^{y_c}\}$ describes the label distribution for $\mathbf{x}_i$, where $c$ is the number of labels, and $d_{\mathbf{x}_i}^{y_l}$ signifies the degree to which label $y_l$ is applicable to $\mathbf{x}_i$, with the constraint that $\sum_{j=1}^{c} d_{\mathbf{x}_i}^{y_j} = 1$. The objective is to train a Deep Neural Network (DNN), symbolized as $f(\mathbf{x}; \theta)$, to accurately predict these label distributions. Each label's output $f_j$ from the model is normalized using the Softmax function (Jang et al., 2016) to ensure it forms a valid probability distribution:

$$h(y_j | \mathbf{x}_i; \theta) = \frac{\exp(f_j(\mathbf{x}_i; \theta))}{\sum_q \exp(f_q(\mathbf{x}_i; \theta))}, \tag{1}$$

where $f_j(\mathbf{x}_i; \theta)$ represents the DNN's raw output for label $y_j$ and instance $\mathbf{x}_i$. And $h(y_j | \mathbf{x}_i; \theta)$ represents the importance degree of the label $y_j$ for $\mathbf{x}_i$. The denominator aggregates the exponential outputs for all potential labels, guaranteeing that the sum of outputs for each instance equals 1 (Gao et al., 2017).

### 3.2 THE SUPERVISED LOSS

In Label Distribution Learning (LDL), we transition from using traditional binary cross-entropy loss, common in multi-label learning (Hershey & Olsen, 2007), to employing Kullback-Leibler (KL) divergence as our loss function. This shift is necessary because LDL predicts continuous real-valued vectors instead of discrete binary outcomes. The KL divergence (Hershey & Olsen, 2007) effectively measures the difference between the actual and the predicted label distributions. The formula for the supervised loss is defined as:

$$\mathcal{L}_s = \frac{1}{n} \sum_{i=1}^{n} \sum_{j=1}^{c} d_{\mathbf{x}_i}^{y_j} \ln\left(\frac{d_{\mathbf{x}_i}^{y_j}}{h(y_j \mid \mathrm{Aug}_w(\mathbf{x}_i); \theta)}\right), \tag{2}$$

here, $\mathrm{Aug}_w(\mathbf{x}_i)$ indicates the weak augmentation (Sohn et al., 2020) applied to the $i$-th sample, and $h(y_j \mid \mathrm{Aug}_w(\mathbf{x}_i); \theta)$ denotes the DNN's predicted label description degree for $y_j$. Employing data augmentation promotes sample diversity, which helps the model learn more generalized features rather than overfitting to specific noise within the training data. This approach not only minimizes the risk of overfitting but also enhances the model's performance on unseen data.

### 3.3 THE UNSUPERVISED CONSISTENCY LOSS

In the realm of SSLDL, a principal challenge is to effectively harness both labeled and a substantial volume of unlabeled data. A prominent strategy that addresses this challenge is consistency regularization, a technique inspired by recent innovations in SSL (Jiang et al., 2022) (Sohn et al., 2020) (Yang et al., 2022) (Zhang et al., 2021). The core idea of this approach is to maintain the consistency of classifier outputs for various augmentations of the same unlabeled instance, thereby ensuring the reliability of label distribution predictions.

To enhance the stability of predictions and maximize the utility of unlabeled data, we adopt an ensemble learning-based approach (Zhou & Zhou, 2021). Instead of relying solely on high-confidence predictions, this method averages the outputs from multiple weakly augmented versions of the same unlabeled image (Sohn et al., 2020), creating what we term the pseudo-label distribution (PLD) for each instance, denoted as $\mathbf{p}_i$. Specifically, for an unlabeled image $\mathbf{x}$, the model produces probability distributions for each of its $H$ weakly augmented versions $\mathrm{Aug}_w(\mathbf{x})$. The PLD is then obtained by averaging these distributions and applying the softmax function to smooth out discrepancies caused by random variations in the data augmentation process: $\mathbf{p}_i = \mathrm{softmax}\left(\frac{1}{H}\sum_{k=1}^{H} p(y|\mathrm{Aug}_w(\mathbf{x})_k; \theta)\right)$.

We quantify the unsupervised consistency loss, $\mathcal{L}_{uc}$, by comparing the PLD against the predictions for strongly augmented versions of the same instances (Sohn et al., 2020). The loss function is mathematically represented as follows:

$$\mathcal{L}_{uc} = \frac{1}{m}\sum_{u=1}^{m}\sum_{j=1}^{c}\left(p_{\mathbf{x}_u}^{y_j}\ln\left(\frac{p_{\mathbf{x}_u}^{y_j}}{h\left(y_j \mid \mathrm{Aug}_s(\mathbf{x}_u); \theta\right)}\right)\right), \tag{3}$$

where $h\left(y_j \mid \mathrm{Aug}_s(\mathbf{x}_u); \theta\right)$ denotes the prediction for label $y_j$ following strong augmentation (Sohn et al., 2020). By integrating this loss function, our model is guided to exploit the inherent structure of the data, fostering learning even in the absence of explicit labels.

### 3.4 THE PAIRWISE RELEVANCE RANKING LOSS

The supervised loss and the unsupervised consistency loss both treat the predicted results and ground-truth (or PLD) as multiple independent prediction tasks, thereby overlooking the inter-label correlation (Xu & Zhou, 2017), which may lead to a decrease in performance. In LDL, a sample is assigned multiple label description degree, and these description degree are often not completely independent of each other (Jia et al., 2018). The correlation between the description degrees can be either positive or negative. For example, if an image $\mathbf{x}$ has a label distribution of $d_{\mathbf{x}}^{y_1} = 0.6$ and $d_{\mathbf{x}}^{y_2} = 0.2$, we consider labels $y_1$ and $y_2$ to be negatively correlated. Similarly, if the labels have a distribution of $d_{\mathbf{x}}^{y_1} = 0.4$ and $d_{\mathbf{x}}^{y_2} = 0.4$, we consider labels $y_1$ and $y_2$ to be positively correlated. This pairwise ranking relationship implicitly expresses the label correlation between label distributions. To tackle this challenge, we introduce a pairwise relevance ranking (PRR) loss $\mathcal{L}_{PRR}$

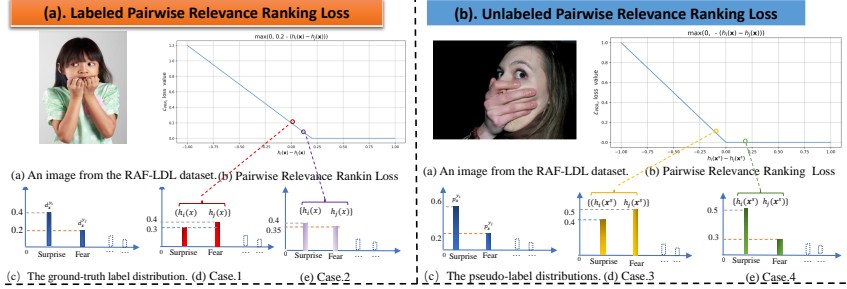

Figure 2: An example to illustrate the $\mathcal{L}_{PRR}$ loss.

to align this inherent semantic structure. For labeled data, we aim for a strict alignment between the ranking of predicted label distributions and the ground-truth. This means that we not only need to align the ranking relationships between label descriptions but also maintain the margin with the ground-truth. Additionally, for certain "close" description degrees, studying their ranking is not meaningful. For instance, consider a scenario where the label description degrees $d_{\mathbf{x}}^{y_i}$ and $d_{\mathbf{x}}^{y_k}$ are 0.32 and 0.33, respectively. The negligible discrepancy between these two values could be attributed to variations in annotation. Consequently, we opt not to adjust their ranking order to account for such minor differences, which may not reflect actual dissimilarities in label importance. Simplifying our notation, let $h_j(\mathbf{x}_i)$ represent the predicted degree of relevance for the j-th label after applying a weak augmentation $\mathrm{Aug}_w$ to the i-th instance. The $\mathcal{L}_{PRR_L}$ loss is then defined as follows:

$$\mathcal{L}_{PRR_L} = \sum_{1 < j < k < q} I(d_{\mathbf{x}_i}^{y_j}, d_{\mathbf{x}_i}^{y_k}) \cdot \max(0, \delta - (h_j(\mathbf{x}_i) - h_k(\mathbf{x}_i)))$$
$$+ I(d_{\mathbf{x}_i}^{y_k}, d_{\mathbf{x}_i}^{y_j}) \cdot \max(0, \delta - (h_k(\mathbf{x}_i) - h_j(\mathbf{x}_i))), \tag{4}$$

Fig. 2, Part (a), presents an image from the RAF-LDL dataset and its label distribution, illustrating the application of the $\mathcal{L}_{PRR_L}$ loss. Here, $\delta = d_{\mathbf{x}_i}^{y_k} - d_{\mathbf{x}_i}^{y_j}$ and the function $I(d_{\mathbf{x}_i}^{y_j}, d_{\mathbf{x}_i}^{y_k})$ is an indicator that outputs 1 if the first label's degree is greater than the second's and their difference is significant, i.e., $d_{\mathbf{x}_i}^{y_j} > d_{\mathbf{x}_i}^{y_k}$ and $|d_{\mathbf{x}_i}^{y_j} - d_{\mathbf{x}_i}^{y_k}| > t$. The loss comes into play in two key scenarios: Case 1, when the model's predicted ranking of labels is incorrect, and Case 2, when the ranking is correct but the margin does not align with the ground truth. Both cases indicate opportunities for the model to learn and adjust its predictions.

In the unsupervised component of our model, we confront the absence of ground-truth labels by employing pseudo-label distributions (PLDs) as a stand-in during training. Recognizing that PLDs may not always be precise, we focus on aligning the predicted pairwise relevance rankings of label descriptions to mitigate the potential for overfitting and to correct inaccuracies inherent in SSL. We define the unsupervised pairwise relevance ranking loss, $\mathcal{L}_{PRR_u}$, where $h_j(\mathbf{x}_i^s)$ denotes the predicted relevance of the j-th label after strong augmentation, $\text{Aug}_s$, is applied to the i-th instance. The loss function is as follows:

$$
\begin{aligned}
\mathcal{L}_{PRR_u} = \sum_{1 < j < k < q} & I(p_{\mathbf{x}_i}^{y_j}, p_{\mathbf{x}_i}^{y_k}) \cdot \max(0, -(h_j(\mathbf{x}_i^s) - h_k(\mathbf{x}_i^s))) \\
& + I(p_{\mathbf{x}_i}^{y_k}, p_{\mathbf{x}_i}^{y_j}) \cdot \max(0, -(h_k(\mathbf{x}_i^s) - h_j(\mathbf{x}_i^s))),
\end{aligned}
\tag{5}
$$

where the indicator function, $I(p_{\mathbf{x}_i}^{y_j}, p_{\mathbf{x}_i}^{y_k})$, outputs 1 if the pseudo-label of one label is greater than the other and their difference is substantial, specifically when $p_{\mathbf{x}_i}^{y_j} > p_{\mathbf{x}_i}^{y_k}$ and the difference $|p_{\mathbf{x}_i}^{y_j} - p_{\mathbf{x}_i}^{y_k}|$ exceeds a threshold $t$; otherwise, it outputs 0. This loss addresses the scenario where the model's ranking of label predictions is inaccurate, as illustrated in Fig. 2, Part (b). Here, we see an image from the RAF-LDL dataset and its associated pseudo-label distribution. For example, when the PLD for surprise ($p_{\mathbf{x}}^{y_i}$) is 0.6 and for fear ($p_{\mathbf{x}}^{y_j}$) is 0.2, the $\mathcal{L}_{PRR}$ loss is activated as $\max(0, -(h_i(\mathbf{x}) - h_j(\mathbf{x})))$, emphasizing the need for the model to correct the predicted rankings to reflect the pseudo-labels more accurately.

Overall, the RankMatch algorithm utilizes this dual-phase training strategy to effectively differentiate between labeled and unlabeled data, continuously refining the model's learning process. The combined application of supervised and unsupervised ranking losses under the PRR framework is modulated by a lambda coefficient ($\lambda$), balancing their contributions. Consequently, the total loss is computed as: $\text{loss} = \mathcal{L}_s + \mathcal{L}_{uc} + \lambda(\mathcal{L}_{PRR_L} + \mathcal{L}_{PRR_u})$ ensuring the model effectively learns from both labeled and unlabeled datasets in a structured manner. The pseudo-code of the RankMatch algorithm can be found in Appendix A.

## 4 THEORETICAL ANALYSIS

**Generalization Bound:** In this section, we establish a theoretical foundation for our RankMatch algorithm within the realm of Semi-Supervised Label Distribution Learning (SSLDL) by defining a generalization bound. Initially, we define the true risk associated with the classification model $f(x; \theta)$:

$$
R(f) = \mathbb{E}_{(x,y)}[L(f(\mathbf{x}), \mathbf{d})].
$$

Our objective is to construct a robust classification model by reducing the empirical risk $\hat{R}(f) = \hat{R}_L(f) + \hat{R}_U(f)$, where $\hat{R}_L(f)$ pertains to the empirical risk associated with the labeled data $L_L(f(\mathbf{x}), \mathbf{d})$ and $\hat{R}_U(f)$ pertains to that of the unlabeled data $L_U(f(\mathbf{x}), \mathbf{d})$:

$$
\hat{R}_L(f) = \frac{1}{n} \sum_{i=1}^{n} L(f(\mathbf{x}_i), \mathbf{d}_i), \quad \hat{R}_U(f) = \frac{1}{m} \sum_{j=1}^{m} L_U(f(\mathbf{x}_j), \mathbf{d}_j).
$$

During model training, direct optimization of $\hat{R}_U(f)$ is impractical as the actual labels of the unlabeled data are unknown. Instead, the model is trained using $\hat{R}'_U(f) = \frac{1}{m} \sum_{j=1}^{m} L_U(f(\mathbf{x}_j), \hat{\mathbf{d}}_j)$, where $\hat{\mathbf{d}}_j$ is the estimated label distribution of the instance $\mathbf{x}_j$.

Let $L_k(f(\mathbf{x})) = d_{\mathbf{x}}^{y_k} \ln\left(\frac{d_{\mathbf{x}}^{y_k}}{h(y_k | \text{Aug}_w(\mathbf{x}))}\right)$ denote the loss for label k, with $L_E$ representing any chosen (but not necessarily optimal) Lipschitz constant for L. Let $R_N(\mathcal{F})$ denote the expected Rademacher

complexity for the function class $\mathcal{F}$ over $N = m + n$ training samples. Assume $\hat{f}$ minimizes the empirical risk, and $f^*$ is the actual risk minimizer. We establish the following theorem to provide a bound on the generalization error. The proof can be find in Appendix D.

**Theorem 1.** *Assuming $\ell(\cdot)$ is limited by $B$, and for some $\epsilon > 0$, if $\sum_{j=1}^{m} | \mathbb{I}(f_k(\mathbf{x}_j)) - \mathbb{I}\left(d_{\mathbf{x}_j}^{y_k}\right) |$ $/m \le \epsilon$ across all $k \in [q]$, for any $\delta > 0$, with a minimum likelihood of $1 - \delta$, we have*

$$R(\hat{f}) - R(f^*) \le 2qB\epsilon + 4qL_E R_N(\mathcal{F}) + 2qB\sqrt{\frac{\log \frac{2}{\delta}}{2N}}.$$

Theorem 2 indicates that $\hat{f}$'s generalization effectiveness primarily hinges on the pseudo-labeling error $\epsilon$ and the aggregate number of training instances $N$. Notably, reducing $\epsilon$ tends to enhance model generalization. Given its inherent robustness and empirical validation, $\hat{f}$ is expected to perform well in real-world settings. Moreover, as $N$ increases indefinitely and $\epsilon$ approaches zero, Theorem 2 confirms that $\hat{f}$ will asymptotically align with the true minimizer $f^\star$.

## 4.1 EXPERIMENTS

### 4.1.1 EXPERIMENTAL CONFIGURATIONS

**Experimental Datasets** We evaluate our approach using four real-world datasets [1]. Briefly:

- **Twitter-LDL** (Yang et al., 2017): Comprises 10,045 images annotated for eight emotions, collected via emotion-specific keyword searches on Twitter.
- **Flickr-LDL** (Yang et al., 2017): A Flickr subset of 10,700 images, labeled for eight emotions by 11 annotators, gathered using adjective-noun pairs.
- **Emotion6** (Peng et al., 2015): Contains 1,980 images sourced from Flickr using keywords for six emotions, each represented in a probability distribution.
- **RAF-LDL** (Li & Deng, 2019): Consists of around 5,000 multi-label distribution facial images, annotated to capture a wide array of emotional expressions.

**Comparing Methods** To evaluate the effectiveness of our proposed RankMatch method, we benchmark it against four distinct groups of algorithms:

- **Deep Learning SSLDL Algorithms:** We introduce two novel algorithms, FixMatch-LDL (Sohn et al., 2020) and MixMatch-LDL (Berthelot et al., 2019), designed to bridge the gap in open-source semi-supervised LDL (SSLDL) approaches within deep learning frameworks.
- **Dual-Network SSLDL Algorithm:** We present and evaluate our own GCT-LDL (Chen et al., 2021), a dual-network SSLDL approach that we developed, which leverages mutual supervision of unlabeled data between two independent networks for enhanced learning.
- **Traditional SSLDL Algorithm:** The traditional SA-LDL (Hou et al., 2017) algorithm, originally for tabular data, is adapted for image datasets through necessary feature engineering, detailed in Appendix A.
- **Existing LDL Algorithms:** Comparisons are also made with state-of-the-art LDL algorithms including Adam-LDL-SCL (Jia et al., 2019), sLDLF (Shen et al., 2017), DF-LDL (González et al., 2021), and LDL-LRR (Jia et al., 2021a), highlighting their potential limitations in SSLDL contexts.

All algorithm configurations and additional methodological details are provided in Appendix A.

**Evaluation Metrics**: In evaluating LDL methods, we employ six distinct metrics (Geng, 2016): Chebyshev, Clark, and Canberra distances, along with Kullback-Leibler divergence, where lower values are preferable, and Intersection and Cosine similarities, where higher values indicate better performance. Details of the evaluation metrics are provided in the Appendix B.

---

[1]The dataset's author has made the dataset publicly available at the following link: http://cv.nankai.edu.cn/projects/SentiLDL. Detailed of these datasets are provided in Appendix A.

### 4.1.2 COMPARATIVE EXPERIMENT ANALYSIS

Table 1: Performance metrics of RankMatch and benchmark semi-supervised label distribution learning algorithms on Emotion6, Flickr, RAF, and Twitter datasets. Results are evaluated at different training sample proportions: 10%, 20%, and 40%. Metrics are shown for Canberra, Clark, KL and Chebyshev distances, with lower scores denoting superior model performance.

| | Method | Emotion6 10% | 20% | 40% | Flickr-LDL 10% | 20% | 40% | Twitter-LDL 10% | 20% | 40% | RAF-LDL 10% | 20% | 40% |
|---|---|---|---|---|---|---|---|---|---|---|---|---|---|
| Can.↓ | Rankmatch | **3.3902** | **3.3176** | 3.2504 | **4.4060** | **3.9964** | **3.9013** | **3.7370** | **3.6962** | **3.2913** | 3.0178 | 2.9358 | 2.8341 |
| | fixmatch-LDL | 3.5080 | 3.5680 | 3.6050 | 5.5570 | 5.5310 | 5.4350 | 6.1750 | 6.0060 | 5.8340 | 3.1220 | 3.0920 | 3.0770 |
| | mixmatch-LDL | 3.6080 | 3.4860 | 3.4880 | 5.6450 | 5.5026 | 5.5750 | 6.3530 | 6.2489 | 6.2960 | 3.1580 | 3.1111 | 3.0630 |
| | GCT-LDL | 3.5980 | 3.5490 | 3.6410 | 5.5860 | 5.5872 | 5.5260 | 6.3010 | 6.3078 | 6.2380 | 3.1920 | 3.1260 | 3.1470 |
| | SALDL | 3.4836 | 3.3737 | **3.1931** | 5.4612 | 4.7789 | 4.8199 | 5.0380 | 4.0868 | 4.0742 | 3.1947 | 3.1415 | 3.0527 |
| | sLDLF | 4.4164 | 4.3398 | 4.1322 | 6.2280 | 6.1238 | 6.2589 | 5.3084 | 6.0008 | 6.1910 | 4.0586 | 4.1705 | 4.1189 |
| | DF-LDL | 4.2427 | 4.0717 | 3.7221 | 5.5348 | 5.5549 | 5.5207 | 6.4184 | 6.3120 | 6.2588 | 3.3281 | 3.3865 | 3.3582 |
| | LDL-LRR | 4.6528 | 4.0496 | 3.7719 | 5.6325 | 5.4988 | 5.4319 | 6.4215 | 6.3295 | 6.2905 | 3.8677 | 4.0116 | 4.1890 |
| | Adam-LDL-SCL | 4.0815 | 4.1128 | 4.1204 | 6.1634 | 5.9889 | 5.6508 | 6.5220 | 6.4081 | 6.3575 | 3.0891 | 3.0242 | 2.9912 |
| Cla.↓ | Rankmatch | **1.5298** | **1.5050** | **1.4834** | **1.8189** | **1.7051** | **1.6737** | **1.6480** | **1.6190** | **1.5138** | **1.4506** | **1.4190** | **1.3843** |
| | fixmatch-LDL | 1.5950 | 1.6230 | 1.6390 | 2.2220 | 2.2110 | 2.1910 | 2.3830 | 2.3310 | 2.2820 | 1.5130 | 1.5060 | 1.5050 |
| | mixmatch-LDL | 1.6240 | 1.5810 | 1.5840 | 2.2330 | 2.1996 | 2.2160 | 2.4280 | 2.4034 | 2.4150 | 1.5150 | 1.5020 | 1.4870 |
| | GCT-LDL | 1.6090 | 1.6050 | 1.6390 | 2.2200 | 2.2238 | 2.2080 | 2.4170 | 2.4216 | 2.4060 | 1.5350 | 1.5170 | 1.5290 |
| | SALDL | 1.6019 | 1.5751 | 1.5100 | 2.1967 | 2.0369 | 2.0446 | 2.1288 | 1.8938 | 1.8964 | 1.5445 | 1.5288 | 1.5035 |
| | sLDLF | 1.8922 | 1.8566 | 1.8049 | 2.3722 | 2.3436 | 2.3761 | 2.1480 | 2.3384 | 2.3746 | 1.9300 | 1.9645 | 1.9750 |
| | DF-LDL | 1.8217 | 1.7746 | 1.6781 | 2.2253 | 2.2072 | 2.1992 | 2.4313 | 2.4108 | 2.4033 | 1.6071 | 1.6229 | 1.6138 |
| | LDL-LRR | 1.9899 | 1.7745 | 1.6953 | 2.2285 | 2.2026 | 2.1919 | 2.4429 | 2.4223 | 2.4121 | 1.7907 | 1.8298 | 1.8919 |
| | Adam-LDL-SCL | 1.7851 | 1.7976 | 1.8014 | 2.3534 | 2.3093 | 2.2312 | 2.4639 | 2.4324 | 2.4160 | 1.5134 | 1.4980 | 1.4905 |
| Int.↑ | Rankmatch | **0.6735** | **0.6832** | **0.6940** | **0.6921** | **0.7073** | **0.7151** | **0.7036** | **0.7190** | **0.7316** | 0.6551 | **0.6813** | **0.7044** |
| | fixmatch-LDL | 0.6638 | 0.6797 | 0.6916 | 0.6857 | 0.7042 | 0.7119 | 0.7009 | 0.7147 | 0.7283 | **0.6570** | 0.6760 | 0.6987 |
| | mixmatch-LDL | 0.6372 | 0.6418 | 0.6496 | 0.6639 | 0.6686 | 0.6831 | 0.6819 | 0.6806 | 0.6986 | 0.6133 | 0.6381 | 0.6534 |
| | GCT-LDL | 0.6116 | 0.6602 | 0.6770 | 0.6639 | 0.6879 | 0.6863 | 0.6787 | 0.7018 | 0.7102 | 0.6669 | 0.6669 | 0.6910 |
| | SALDL | 0.6457 | 0.6612 | 0.6723 | 0.5559 | 0.5108 | 0.5091 | 0.6632 | 0.5724 | 0.5687 | 0.6298 | 0.6504 | 0.6708 |
| | sLDLF | 0.5935 | 0.5861 | 0.6162 | 0.4813 | 0.4750 | 0.4616 | 0.6487 | 0.5652 | 0.5336 | 0.2433 | 0.2315 | 0.2199 |
| | DF-LDL | 0.5057 | 0.5461 | 0.6353 | 0.4173 | 0.4176 | 0.4169 | 0.3541 | 0.3536 | 0.3505 | 0.7022 | 0.7083 | 0.7085 |
| | LDL-LRR | 0.3721 | 0.6213 | 0.6626 | 0.5322 | 0.5519 | 0.5600 | 0.5746 | 0.5904 | 0.5979 | 0.5649 | 0.5389 | 0.4411 |
| | Adam-LDL-SCL | 0.3409 | 0.5627 | 0.6040 | 0.4724 | 0.3933 | 0.4628 | 0.5488 | 0.5828 | 0.5200 | 0.6177 | 0.5768 | 0.4843 |
| Cos.↑ | Rankmatch | **0.8121** | **0.8257** | **0.8331** | **0.8489** | **0.8614** | **0.8679** | **0.8544** | **0.8698** | **0.8790** | 0.7901 | **0.8140** | 0.8375 |
| | fixmatch-LDL | 0.8079 | 0.8200 | 0.8312 | 0.8487 | 0.8573 | 0.8673 | 0.8517 | 0.8647 | 0.8758 | 0.7881 | 0.8123 | 0.8311 |
| | mixmatch-LDL | 0.7585 | 0.7863 | 0.7901 | 0.7888 | 0.8381 | 0.8468 | 0.8463 | 0.8552 | 0.8602 | 0.7536 | 0.7680 | 0.7820 |
| | GCT-LDL | 0.7530 | 0.8017 | 0.8134 | 0.8313 | 0.8508 | 0.8531 | 0.8499 | 0.8587 | 0.8716 | 0.7660 | 0.7977 | 0.8181 |
| | SALDL | 0.7784 | 0.7874 | 0.7981 | 0.7361 | 0.6643 | 0.6624 | 0.8479 | 0.7612 | 0.7615 | 0.7711 | 0.7938 | 0.8135 |
| | sLDLF | 0.7037 | 0.6980 | 0.7350 | 0.6276 | 0.6066 | 0.5897 | 0.8002 | 0.7454 | 0.6988 | 0.3262 | 0.3506 | 0.3459 |
| | DF-LDL | 0.6035 | 0.6470 | 0.7689 | 0.5436 | 0.5539 | 0.5569 | 0.5069 | 0.5233 | 0.5209 | 0.8427 | 0.8492 | 0.8470 |
| | LDL-LRR | 0.4604 | 0.7362 | 0.7905 | 0.7020 | 0.7316 | 0.7399 | 0.7767 | 0.8027 | 0.8125 | 0.7253 | 0.6938 | 0.5757 |
| | Adam-LDL-SCL | 0.4311 | 0.6670 | 0.7144 | 0.6104 | 0.4888 | 0.6166 | 0.7163 | 0.7661 | 0.7403 | 0.7717 | 0.7337 | 0.6191 |

We employed a range of labeled data proportions (10%, 20%, and 40%) to simulate varying levels of label availability, a critical factor in semi-supervised learning scenarios. Our evaluation metrics included Canberra, Clark, Intersection and Cosine[2]. The experiments are presented in Table. 1. Furthermore, we train using full supervision information on RankMatch, and the experimental results are presented in Table .2.from that we can draw the following conclusions

- RankMatch consistently achieves top performance across all datasets (Emotion6, Flickr, Twitter, RAF) and metrics (Intersection, Cosine, KL, Chebyshev).

- As a deep learning-based method, RankMatch substantially outperforms traditional models such as GCT-LDL and fixmatch-LDL. It excels by leveraging complex, hierarchical features from data, which traditional models often miss due to reliance on simpler features and assumptions.

- Despite the enhancements in SSLDL algorithms like fixmatch-LDL and mixmatch-LDL through deep learning, RankMatch surpasses them in all metrics. Its advantage stems from the effective use of deep learning techniques combined with an understanding of label relationships. This is vital in LDL tasks like emotion recognition, where the accurate modeling of emotional intensity and distribution is crucial. RankMatch's ability to account for these relationships enables it to deliver more precise and relevant predictions than models processing labels independently, enhancing both practicality and accuracy in applications.

- Analyzing Table. 2, RankMatch's performance improves significantly as the percentage of training samples increases, closely approaching fully supervised outcomes by using just 40% of the data. This demonstrates RankMatch's effectiveness as a semi-supervised learning

---

[2]The results utilizing KL and Chebyshev are detailed in the Appendix C.

method, efficiently utilizing less labeled data to achieve near-complete performance, which highlights its potential in applications with limited labeled resources.

Table 2: RankMatch's performance across varying training sample sizes is measured by six metrics. During the 100% data training process, no unsupervised components are included.

| | Emotion6 | | | | Flickr-LDL | | | | Twitter-LDL | | | | Raf-LDL | | | |
|---|---|---|---|---|---|---|---|---|---|---|---|---|---|---|---|---|
| | 10% | 20% | 40% | 100% | 10% | 20% | 40% | 100% | 10% | 20% | 40% | 100% | 10% | 20% | 40% | 100% |
| Can. ↓ | 3.3902 | 3.3176 | 3.2504 | 3.203 | 4.4060 | 3.9964 | 3.9013 | 3.625 | 3.7370 | 3.6962 | 3.2913 | 2.902 | 3.0178 | 2.9358 | 2.8341 | 2.794 |
| Cla.↓ | 1.5298 | 1.5050 | 1.4834 | 1.467 | 1.8189 | 1.7051 | 1.6737 | 1.595 | 1.6480 | 1.6190 | 1.5138 | 1.395 | 1.4506 | 1.4190 | 1.3843 | 1.366 |
| Cos. ↑ | 0.8121 | 0.8257 | 0.8331 | 0.845 | 0.8489 | 0.8614 | 0.8679 | 0.8694 | 0.8544 | 0.8698 | 0.8790 | 0.8827 | 0.7901 | 0.8140 | 0.8375 | 0.8478 |
| Int. ↑ | 0.6735 | 0.6832 | 0.6940 | 0.7055 | 0.6735 | 0.6832 | 0.6940 | 0.7176 | 0.7036 | 0.7190 | 0.7316 | 0.7411 | 0.6551 | 0.6813 | 0.7044 | 0.7188 |

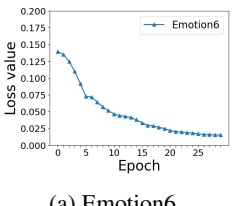 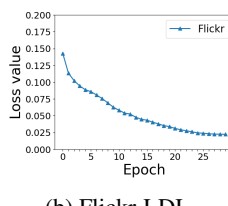 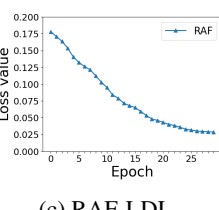 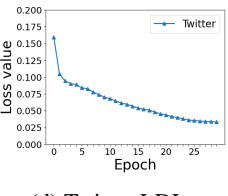

(a) Emotion6      (b) Flickr-LDL      (c) RAF-LDL      (d) Twitter-LDL

Figure 3: The convergence curve on Emotion6, FLickr-LDL, RAF-ML, and Twitter-LDL.

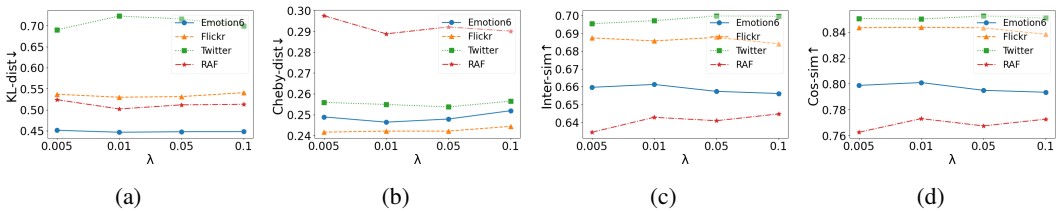

(a)      (b)      (c)      (d)

Figure 4: Parameter Sensitivity Analysis on (a) Emotion6, (b) Flickr-LDL, (c) RAF-LDL and (d) Twitter-LDL.

### 4.1.3 FURTHER ANALYSIS

**Ablation Study** Our ablation study analyzed the impact of PRR loss and unsupervised consistency loss on the performance of RankMatch. Initially, the model was pre-trained with only 10% of labeled data to establish a baseline. This phase highlighted the model's ability to utilize minimal data effectively.

Next, unsupervised consistency loss was applied to enhance learning from unlabeled data. In the final phase, PRR loss was introduced, leveraging the same 10% labeled data to refine the model further with supervised ranking loss. Ablation experiment results are shown in Table. 7. From this, we can draw the following conclusions

- The integration of unsupervised consistency loss markedly improves RankMatch's performance across datasets, as observed in the ablation results. This confirms the effectiveness of using unsupervised data to enhance model accuracy.

- The incorporation of pairwise relevance ranking (PRR) loss significantly boosts performance, particularly in scenarios where it surpasses the baseline. This improvement demonstrates the

Table 3: Ablation Results on Emotion6 and Twitter-LDL.

| | | Che.↓ | Cla.↓ | Can.↓ | KL↓ | Cos.↑ | Int.↑ |
|---|---|---|---|---|---|---|---|
| Emotion6 | pretrain | 0.2504 | 1.6524 | 3.6893 | 0.4642 | 0.793 | 0.6557 |
| | pretrain + consistency | 0.2362(5.7%↑) | 1.623(1.8%↑) | 3.5537(3.7%↑) | 0.4273(7.9%↑) | 0.8216(3.6%↑) | 0.6789(3.5%↑) |
| | pretrain + consistency+PRR loss | 0.2186(7.5%↑) | 1.6028(1.2%↑) | 3.4761(2.2%↑) | 0.3776(11.6%↑) | 0.8349(1.6%↑) | 0.6982(2.8%↑) |
| Twitter-LDL | pretrain | 0.2538 | 2.463 | 6.4139 | 0.7908 | 0.8502 | 0.701 |
| | pretrain + consistency | 0.2442(3.8%↑) | 1.9025(22.8%↑) | 4.6251(27.9%↑) | 0.6854(13.3%↑) | 0.8608(1.2%↑) | 0.7242(3.3%↑) |
| | pretrain + consistency+PRR loss | 0.2262(7.4%↑) | 1.7382(8.6%↑) | 4.0088(13.3%↑) | 0.6232(13.3%↑) | 0.8799(2.2%↑) | 0.7369(1.8%↑) |

PRR loss's critical role in refining label discrimination within the semi-supervised learning framework.

**Convergence Analysis** The convergence behavior of the RankMatch algorithm is evaluated in Fig. 3, showing varied learning dynamics across datasets. For Emotion6 and Flickr-LDL, rapid initial loss declines indicate swift learning and quick stabilization, suggesting efficient adaptation. Conversely, RAF-LDL and Twitter-LDL exhibit slower, steadier loss reductions, highlighting methodical learning. Overall, consistent loss improvement across all datasets demonstrates RankMatch's effective optimization, enhancing predictive accuracy over training.

**Parameter Sensitivity Analysis** Based on the results shown in Figure. 4 [3], we analyze the impact of parameter $\lambda$ on RankMatch's performance across Emotion6, Flickr-LDL, RAF-LDL, and Twitter-LDL datasets. The analysis yields the following insights

- **Stability Across $\lambda$ Values:** RankMatch shows high stability when $\lambda$ ranges from 0.01 to 0.05, with minimal variations in key performance metrics (KL, Chebyshev, Intersection, and Cosine) across all datasets. This range appears to be optimal for $\lambda$, allowing the algorithm to maintain effective performance.

- **Impact of Low $\lambda$ Values:** At $\lambda$ values close to 0.005, performance deteriorates significantly, as seen in the increase in KL for the Emotion6 dataset. This suggests that low $\lambda$ values reduce the effectiveness of the regularization, leading to poorer learning outcomes.

Table 4: Impact of Threshold t on the Performance of the RankMatch.

| Emotion6 | t=0 | t=0.01 | t=0.05 | t=0.1 | t=0.2 | t=0.3 | t=0.4 | t=0.7 | t=1 |
|---|---|---|---|---|---|---|---|---|---|
| Cos.↑ | 0.7912 | 0.7903 | 0.7921 | 0.7964 | 0.7977 | **0.8033** | 0.7967 | 0.7933 | 0.7857 |
| KL↓ | 0.4603 | 0.4577 | 0.4505 | 0.4425 | 0.4491 | **0.4419** | 0.4502 | 0.4603 | 0.4597 |
| RAF-LDL | t=0 | t=0.01 | t=0.05 | t=0.1 | t=0.2 | t=0.3 | t=0.4 | t=0.7 | t=1 |
| Cos.↑ | 0.7512 | 0.7533 | 0.7671 | 0.771 | 0.7726 | **0.7743** | 0.7642 | 0.7581 | 0.7554 |
| KL↓ | 0.5268 | 0.5224 | 0.522 | 0.5078 | **0.4954** | 0.5114 | 0.5226 | 0.5284 | 0.5339 |

**Impact of Threshold t on Experimental Results** The threshold $t$ in the Pairwise Relevance Ranking (PRR) loss significantly influences RankMatch's sensitivity to label ranking discrepancies. As detailed in Table 4, our experiments across various datasets and metrics lead to two key conclusions about this impact on the algorithm's performance.

- **Optimal Performance Range:** For both datasets, Emotion6 and RAF, RankMatch shows optimal performance when $t$ is set between 0.2 and 0.3. This range yields the lowest scores for both the Cosine and KL divergence metrics, indicating an effective balance in the model's ability to manage the inter-label dynamics. This suggests that a moderate threshold level is crucial for maximizing the utility of the PRR loss.

- **Performance Decline at Extreme Values of $t$:** At the extremes, $t = 0$ and $t = 1$, there is a significant decline in performance. At $t = 0$, the PRR loss component is essentially non-operational, which results in inadequate penalization for misranked labels. Conversely, at $t = 1$, an overly restrictive threshold may limit the model's adaptability, hindering its learning capabilities from the data. This behavior is especially pronounced in the RAF dataset, where performance metrics deteriorate notably at these values.

## 5 CONCLUSION

In this paper, we introduce RankMatch, an innovative semi-supervised label distribution learning (SSLDL) method. RankMatch utilizes a combination of a small amount of labeled data with a substantial quantity of unlabeled data, minimizing the need for extensive manual labeling. It employs an averaging approach inspired by ensemble learning to generate stable pseudo-label distributions and incorporates a novel relevance ranking loss to effectively manage label correlations. We provide a theoretical generalization bound for RankMatch, and our comprehensive experimental results demonstrate its superiority over existing SSLDL approaches in effectively tackling various SSLDL challenges.

---

[3]Extended parameter results are presented in Appendix C.

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

# A  APPENDIX

# B  APPENDIX (DETAIL OF THE DATASET, AND COMPARISON SETTINGS OF ALGORITHMS.)

---

**Algorithm 1:** The pseudo-code of the RankMatch

---

**Input:** Set of labeled examples and their label distribution $\mathcal{D}_L = \{(x_i, d_i)|i \leq n\}$, Set of unlabeled data $\mathcal{D}_U = \{x_j | j \leq m\}$, Model pretrained on labeled dataset $f_p(\cdot\,;\theta_p)$, number of train epochs $N$, number of different weakly augmentations $K$

**Output:** model $f(\cdot\,;\theta)$

1  Init a model $f(\cdot\,;\theta)$;
2  **for** $i = 1, \ldots, N$ **do**
3      **for** each $x$ in $\{\mathcal{D}_L \cup \mathcal{D}_U\}$ **do**
4          **if** $x \in \mathcal{D}_\mathcal{L}$ **then**            // Processing for labeled data
5             $h_l = \text{Softmax}(f(\text{Aug}_s(x);\theta))$;
6             $\text{Loss} = \mathcal{L}_s(h_l, d) + \lambda\mathcal{L}_{PRR_L}(h_l, d)$;
7          **else if** $x \in \mathcal{D}_\mathcal{U}$ **then**      // Processing for unlabeled data
           // Average outputs from K weakly augmented unlabeled
           image $p = \frac{1}{K}\sum_{j=1}^{K}\text{Softmax}(f_p(\text{Aug}_{wj}(x);\theta_p))$;
8             $h_U = \text{Softmax}(f(\text{Aug}_w(x);\theta))$;
9             $\text{Loss} = \mathcal{L}_{uc}(h_u, p) + \lambda\mathcal{L}_{PRR_u}(h_u, p)$;
10         Update $\theta$ via $\min_\theta \text{Loss}$;

---

**Experimental Datasets** :In this paper, we validate our approach using four distinct real-world datasets [4]. The details of these datasets are as follows:

Twitter-LDL : A large-scale Visual Sentiment Distribution dataset was constructed from Twitter, encompassing eight distinct emotions Amusement, Anger, Awe, Contentment, Disgust, Excitement, Fear, Sadness. Approximately 30,000 images were collected by searching various emotional keywords, such as "sadness," "heartbreak," and "grief." Subsequently, eight annotators were hired to label this dataset. The resulting Twitter LDL dataset comprises 10,045 images.

Flickr-LDL : A subset of the Flickr dataset , unlike other datasets that searched for images using emotional terms, the Flickr dataset collected 1,200 pairs of adjective-noun pairs, resulting in 500,000 images. We employed 11 annotators to label this subset with tags for eight common emotions. In the end, the Flickr LDL was created, containing 10,700 images, with roughly equal quantities for each class.

Emotion6 : Emotion6: We collected 1,980 images from Flickr using six category keywords and synonyms as search terms for Emotion6. A total of 330 images were collected for each category, and each image was assigned to only one category (dominant emotion). Emotion6 represents the emotions related to each image in the form of a probability distribution, consisting of 7 bins, including Ekman's 6 basic emotions and neutral.

RAF-LDL : RAF-LDL is a multi-label distribution facial expression dataset, comprising approximately 5,000 diverse facial images downloaded from the internet. These images exhibit variations in emotion, subject identity, head pose, lighting conditions, and occlusions. During annotation, 315 well-trained annotators are employed to ensure each image can be annotated enough independent times. And images with multi-peak label distribution are selected out to constitute the RAF-LDL.

**Comparing methods** In order to assess the effectiveness of the proposed approach, we benchmark it against four sets of methods:

1) The first group consists of two deep learning SSLDL algorithms that we introduced, named FixMatch-LDL and MixMatch-LDL. Since there are currently no open-source semi-supervised LDL

---

[4]The dataset's author has made the dataset publicly available at the following link: http://cv.nankai.edu.cn/projects/SentiLDL.

works in deep learning, these two algorithms were developed by us, based on the current most effective two deep learning SSL algorithms.

(a) FixMatch-LDL :Fixmatch-LDL is an adaptation we made based on the classic semi-supervised algorithm fixmatch (Sohn et al., 2020). Specifically, we pre-trained on images using ResNet50, then trained the model with labeled data. Subsequently, we assigned pseudo-label distributions to the unlabeled data, and finally, we aligned the model's strongly augmented output with the pseudo-label distribution. For all datasets, the number of epochs is set as 30 and the batch size is set as 32. We perform all experiments on GeForce RTX 3090 GPUs. The random seed is set to 1 for all experiments.

(b) MixMatch-LDL: Mixmatch is a semi-supervised LDL algorithm designed by us. Specifically, we first use linear interpolation to blend images, creating new samples. Similarly, we generate the label distributions for these new samples. Following this, we train the data using the same training strategy as Mixmatch. It's worth mentioning that producing new samples enhances the model's ability to prevent overfitting. For all datasets, the number of epochs is set as 30 and the batch size is set as 32. We perform all experiments on GeForce RTX 3090 GPUs. The random seed is set to 1 for all experiments.

2) The second group of algorithms is a deep learning SSLDL algorithm based on the dual-network concept, which we named GCT-LDL. The core idea involves mutual supervision of the outputs from two independent networks using unlabeled data. GCT-LDL : Two models utilized two different pretrained initializations of ResNet50 provided by PyTorch (ResNet50-Weights.IMAGENET1K-V1 and ResNet50-Weights.IMAGENET1K-V2). During training, labeled and unlabeled data were mixed. The loss used is the cross-entropy loss, divided into two parts: for labeled data, the loss is calculated directly between the prediction results and the ground truth. For unlabeled data, the loss is calculated between the prediction results of each model and the results of the other model. Hyperparameter settings are the same as those used in other methods.

3) The third group consists of traditional SSLDL algorithms, referred to as SA-LDL (Hou et al., 2017). Since SA-LDL is an SSLDL algorithm designed for tabular data, we needed to perform feature engineering on image data, first, we use ResNet-50 for feature extraction from all datasets, followed by dimensionality reduction to 128 dimensions using PCA. For the remaining settings, we adhere to the defaults as specified in the paper.

4) The fourth category consists of existing LDL algorithms. As there is currently only one open-source SSLDL algorithm, which is SA-LDL (Hou et al., 2017), we compared it with some state-of-the-art LDL algorithms. In this regard, we selected four state-of-the-art LDL algorithms: Adam-LDL-SCL (Jia et al., 2019), sLDLF (Shen et al., 2017), DF-LDL (González et al., 2021), and LDL-LRR (Jia et al., 2021a). These algorithm settings are defaulted to be consistent with those specified in the paper. Additionally, for these algorithms, we directly use labeled data to train the classifier. Then, we use the trained model to assign pseudo-labels to the unlabeled samples. Finally, we use the pseudo-labels to update the model.

**Implementation** Following (Cole et al., 2021), we employ ResNet-50 (He et al., 2016) pre-trained on ImageNet (Krizhevsky et al., 2012) for training the classification model. For training images, we adopt standard flip-and-shift strategy (Sohn et al., 2020) for weak data augmentation, and RandAugment (Cubuk et al., 2020) and Cutout (DeVries & Taylor, 2017) for strong data augmentation. We employ AdamW (You et al., 2019) optimizer and one-cycle policy scheduler (Hannan et al., 2021) to train the model with maximal learning rate of 0.0001. For all datasets, the number of epochs is set as 30 and the batch size is set as 32. Furthermore, we perform exponential moving average (EMA) (Klinker, 2011) for the model parameter $\theta$ with a decay of 0.98. We adjust the parameter $\lambda$ across a range of values, specifically $\{0.005, 0.01, 0.05, 0.1\}$. We perform all experiments on GeForce RTX 3090 GPUs. The random seed is set to 1 for all experiments.

**Implementation** Following (Cole et al., 2021), we employ ResNet-50 (He et al., 2016) pre-trained on ImageNet (Krizhevsky et al., 2012) for training the classification model. For training images, we adopt standard flip-and-shift strategy (Sohn et al., 2020) for weak data augmentation, and RandAugment (Cubuk et al., 2020) and Cutout (DeVries & Taylor, 2017) for strong data augmentation. We employ AdamW (You et al., 2019) optimizer and one-cycle policy scheduler (Hannan et al., 2021) to train the model with maximal learning rate of 0.0001. For all datasets, the number of epochs is set as 30 and the batch size is set as 32. Furthermore, we perform exponential moving average (EMA) (Klinker, 2011) for the model parameter $\theta$ with a decay of 0.98. We adjust the parameter $\lambda$ across a range

of values, specifically $\{0.005, 0.01, 0.05, 0.1\}$. We perform all experiments on GeForce RTX 3090 GPUs. The random seed is set to 1 for all experiments.

## C   APPENDIX (DETAILS OF THE EVALUATION METRICS FOR THE EXPERIMENTS.)

**Evaluation Metrics**: We evaluate LDL algorithms using six metrics: five distance-based (Chebyshev, Clark, Kullback-Leibler, and Canberra) and two similarity-based (Cosine and Intersection). Formulas for these metrics are provided in the appendix. Lower values indicate better performance for distance-based metrics ($\downarrow$), while higher values indicate better performance for similarity-based metrics ($\uparrow$).

Table 5: The distribution distance/similarity measures

| Measure | Formula |
|---------|---------|
| Chebyshev $\downarrow$ | $\mathrm{Dis}_1(\boldsymbol{d}, \hat{\boldsymbol{d}}) = \max_j \left| d_j - \hat{d}_j \right|$ |
| Clark $\downarrow$ | $\mathrm{Dis}_2(\boldsymbol{d}, \hat{\boldsymbol{d}}) = \sqrt{\sum_{j=1}^{c} \frac{\left(d_j - \hat{d}_j\right)^2}{\left(d_j + \hat{d}_j\right)^2}}$ |
| Canberra $\downarrow$ | $\mathrm{Dis}_3(\boldsymbol{d}, \hat{\boldsymbol{d}}) = \sum_{j=1}^{c} \frac{\left| d_j - \hat{d}_j \right|}{d_j + \hat{d}_j}$ |
| Kullback-Leibler $\downarrow$ | $\mathrm{Dis}_4(\boldsymbol{d}, \hat{\boldsymbol{d}}) = \sum_{j=1}^{c} d_j \ln \frac{d_j}{\hat{d}_j}$ |
| Cosine $\uparrow$ | $\mathrm{Sim}_1(\boldsymbol{d}, \hat{\boldsymbol{d}}) = \frac{\sum_{j=1}^{c} d_j \hat{d}_j}{\sqrt{\sum_{j=1}^{c} d_j^2} \sqrt{\sum_{j=1}^{c} \hat{d}_j^2}}$ |
| Intersection $\uparrow$ | $\mathrm{Sim}_2 = \frac{1}{n} \sum_{i=1}^{n} \sum_{j=1}^{c} \min\left(d_{\boldsymbol{x}_i}^{y_j}, \hat{d}_{\boldsymbol{x}_i}^{y_j}\right)$ |

## D APPENDIX (PRESENTATION OF THE REMAINING EXPERIMENTAL RESULTS.)

Table 6: Performance metrics of RankMatch and benchmark semi-supervised label distribution learning algorithms on Emotion6, Flickr, RAF, and Twitter datasets. Results are evaluated at different training sample proportions: 10%, 20%, and 40%. Metrics are shown for Intersection and Cosine distances, with higher scores denoting superior model performance.

| | | Emotion6 | | | Flickr-LDL | | | Twitter-LDL | | | RAF-LDL | | |
|---|---|---|---|---|---|---|---|---|---|---|---|---|---|
| | Method | 10% | 20% | 40% | 10% | 20% | 40% | 10% | 20% | 40% | 10% | 20% | 40% |
| | Rankmatch | 0.4214 | **0.3916** | **0.3896** | 0.4961 | **0.4836** | 0.4800 | **0.2386** | **0.2279** | **0.2241** | 0.2348 | **0.2248** | 0.2191 |
| | fixmatch-LDL | **0.4175** | 0.4095 | 0.4072 | **0.4921** | 0.5054 | 0.4856 | 0.2433 | 0.2305 | 0.2245 | 0.2382 | 0.2272 | **0.2190** |
| | mixmatch-LDL | 0.504 | 0.4504 | 0.4417 | 0.5109 | 0.4768 | 0.4550 | 0.2685 | 0.2565 | 0.2527 | 0.2543 | 0.2446 | 0.2367 |
| KL ↓ | GCT-LDL | 0.5151 | 0.4276 | 0.4201 | 0.5017 | 0.4552 | **0.4449** | 0.2815 | 0.2469 | 0.2391 | 0.2519 | 0.2355 | 0.2329 |
| | SALDL | 0.5885 | 0.5620 | 0.5608 | 0.7567 | 1.7464 | 1.7848 | 0.2595 | 0.2590 | 0.2486 | 0.3388 | 0.3648 | 0.3651 |
| | sLDLF | 0.8382 | 0.8905 | 0.704 | 1.7006 | 1.6464 | 1.4217 | 0.3195 | 0.3263 | 0.3054 | 0.4032 | 0.414 | 0.4152 |
| | DF-LDL | 1.0926 | 0.8023 | 0.4954 | 1.1285 | 1.0817 | 1.0679 | 0.3681 | 0.3377 | 0.2772 | 0.4376 | 0.4419 | 0.4442 |
| | LDL-LRR | 3.2914 | 0.648 | 0.4431 | 0.8502 | 0.7682 | 0.7418 | 0.5708 | 0.2986 | 0.2611 | 0.3506 | 0.3392 | 0.3363 |
| | Adam-LDL-SCL | 0.7687 | 0.8933 | 0.7725 | 1.5907 | 1.1059 | 0.8454 | 0.3002 | 0.3067 | 0.2989 | 0.4171 | 0.3824 | 0.3527 |
| | | Emotion6 | | | Flickr-LDL | | | Twitter-LDL | | | RAF-LDL | | |
| | Method | 10% | 20% | 40% | 10% | 20% | 40% | 10% | 20% | 40% | 10% | 20% | 40% |
| | Rankmatch | 0.6646 | 0.5490 | **0.5227** | **0.454** | **0.4093** | **0.3585** | **0.2505** | **0.237** | **0.2284** | **0.281** | **0.265** | **0.2452** |
| | fixmatch-LDL | 0.8077 | 0.6438 | 0.5703 | 0.4706 | 0.4188 | 0.3812 | 0.2511 | 0.2403 | 0.232 | 0.2798 | 0.2677 | 0.2489 |
| | mixmatch-LDL | 0.5590 | **0.5136** | 0.4904 | 0.5297 | 0.4945 | 0.4612 | 0.2599 | 0.255 | 0.2446 | 0.3141 | 0.302 | 0.2921 |
| Che.↓ | GCT-LDL | **0.5461** | 0.5263 | 0.4757 | 0.5164 | 0.4466 | 0.3987 | 0.2592 | 0.2468 | 0.2365 | 0.2977 | 0.2768 | 0.2611 |
| | SALDL | 0.7364 | 1.8742 | 1.9519 | 0.5919 | 0.4918 | 0.4912 | 0.267 | 0.3434 | 0.3438 | 0.295 | 0.2784 | 0.2679 |
| | sLDLF | 1.1756 | 1.5865 | 1.5537 | 6.2799 | 6.4985 | 7.9414 | 0.3073 | 0.3705 | 0.3963 | 0.741 | 0.7646 | 0.7801 |
| | DF-LDL | 1.4356 | 1.3234 | 1.307 | 0.5124 | 0.4665 | 0.432 | 0.5153 | 0.52 | 0.5293 | 0.256 | 0.2521 | 0.2471 |
| | LDL-LRR | 0.8999 | 0.7729 | 0.7212 | 1.3878 | 1.4658 | 2.2773 | 0.3364 | 0.3206 | 0.3153 | 0.4102 | 0.4332 | 0.5192 |
| | Adam-LDL-SCL | 0.9974 | 0.8017 | 0.7367 | 0.9481 | 1.1863 | 1.7387 | 0.3612 | 0.3333 | 0.3213 | 1.6141 | 1.1399 | 0.9903 |

Table 7: Ablation Results on 2 Datasets.

| | | Che.↓ | Cla.↓ | Can.↓ | KL↓ | Cos.↑ | Int.↑ |
|---|---|---|---|---|---|---|---|
| | pretrain | 0.2411 | 2.2594 | 5.6885 | 0.5371 | 0.8427 | 0.6873 |
| Flickr | pretrain + consistency | 0.2262(6.2%↑) | 2.1131(6.5%↑) | 5.1536(9.4%↑) | 0.5293(1.5%↑) | 0.8633(2.4%↑) | 0.7188(4.6%↑) |
| | pretrain + consistency+PRR loss | 0.2184(3.4%↑) | 2.0158(4.6%↑) | 4.9008(4.9%↑) | 0.5227(1.2%↑) | 0.8714(0.9%↑) | 0.7208(0.3%↑) |
| | | Che.↓ | Cla.↓ | Can.↓ | KL↓ | Cos.↑ | Int.↑ |
| | pretrain | 0.2938 | 1.5412 | 3.206 | 0.5146 | 0.7687 | 0.6411 |
| RAF | pretrain + consistency | 0.255(13.2%↑) | 1.5021(2.5%↑) | 3.1345(2.2%↑) | 0.3699(28.1%↑) | 0.8189(28.1%↑) | 0.7073(10.3%↑) |
| | pretrain + consistency+PRR loss | 0.2341(8.2%↑) | 1.4914(0.7%↑) | 3.0459(2.8%↑) | 0.3464(6.4%↑) | 0.8476(3.5%↑) | 0.7194(1.7%↑) |

Table 8: The impact of different $\lambda$ values on experimental results.

| | $\lambda=0$ | $\lambda=0.005$ | $\lambda=0.01$ | $\lambda=0.1$ | $\lambda=1$ | $\lambda=10$ | $\lambda=100$ | $\lambda=1000$ |
|---|---|---|---|---|---|---|---|---|
| Che.↓ | 0.2696 | 0.2574 | 0.25 | 0.2519 | 0.2587 | 0.2572 | 0.3024 | 0.3073 |
| Int.↑ | 0.6392 | 0.6576 | 0.6586 | 0.6562 | 0.648 | 0.635 | 0.5477 | 0.5353 |
| | $\lambda=0$ | $\lambda=0.005$ | $\lambda=0.01$ | $\lambda=0.1$ | $\lambda=1$ | $\lambda=10$ | $\lambda=100$ | $\lambda=1000$ |
| Che.↓ | 0.3102 | 0.2975 | 0.2909 | 0.2904 | 0.3104 | 0.3462 | 0.3561 | 0.3553 |
| Int.↑ | 0.6202 | 0.6344 | 0.6416 | 0.644 | 0.635 | 0.5859 | 0.5734 | 0.5726 |

## E APPENDIX (THE PROOF PROCESS OF THEOREM 1.)

### E.1 GENERALIZATION BOUND

We study the generalization performance of Rankmatch. Before providing the main results, we first define the true risk with respect to the classification model $f(x; \theta)$:

$$R(f) = \mathbb{E}_{(x,y)}[L(f(\mathbf{x}), \mathbf{d})].$$

Our goal is to learn a good classification model by minimizing the empirical risk $\hat{R}(f) = \hat{R}_L(f) + \hat{R}_U(f)$, where $\hat{R}_L(f)$ and $\hat{R}_U(f)$ are respectively the empirical risk of the labeled loss $L_L(f(\mathbf{x}), \mathbf{d})$ and unlabeled loss $L_U(f(\mathbf{x}), \mathbf{d})$:

$$\hat{R}_L(f) = \frac{1}{n}\sum_{i=1}^{n} L(f(\mathbf{x}_i), \mathbf{d}_i), \quad \hat{R}_U(f) = \frac{1}{m}\sum_{j=1}^{m} L_U(f(\mathbf{x}_j), \mathbf{d}_j).$$

Note that during the training, we cannot train a model directly by optimizing $\hat{R}_U(f)$, since the labels of unlabeled data are inaccessible. Instead, we train the model with $\hat{R}'_U(f) = \frac{1}{m}\sum_{j=1}^{m} L_U(f(\mathbf{x}_j), \hat{\mathbf{d}}_j)$, where $\hat{\mathbf{d}}_j$ represents the pseudo-label vector of the instance $\mathbf{x}_j$.

Let $L_k(f(\mathbf{x})) = d_{\mathbf{x}}^{y_k} \ln\left(\frac{d_{\mathbf{x}}^{y_k}}{h(y_k|\text{Aug}_w(\mathbf{x}))}\right)$ be the loss for the label k, and $L_E$ be any (not necessarily the best) Lipschitz constant of $L$. Let $R_N(\mathcal{F})$ be the expected Rademacher complexity of $\mathcal{F}$ with $N = m + n$ training points. Let $\hat{f}$ be the empirical risk minimizer, where $\mathcal{F}$ is a function class, and $f^*$ be the true minimizer. We derive the following theorem, which provides a generalization error bound for the proposed method.

**Theorem 2.** *Suppose that $\ell(\cdot)$ is bounded by B. For some $\epsilon > 0$, if $\sum_{j=1}^{m} | \mathbb{I}(f_k(\mathbf{x}_j)) - \mathbb{I}\left(d_{\mathbf{x}_j}^{y_k}\right) |$* */m \le \epsilon$ for any $k \in [q]$, for any $\delta > 0$, with probability at least $1 - \delta$, we have*

$$R(\hat{f}) - R(f^*) \le 2qB\epsilon + 4qL_E R_N(\mathcal{F}) + 2qB\sqrt{\frac{\log\frac{2}{\delta}}{2N}}.$$

From Theorem 2, it can be observed that the generalization performance of $\hat{f}$ mainly depends on two factors, i.e., the pseudo-labeling error $\epsilon$ and the number of training examples $N$. Apparently, a smaller pseudo-labeling error $\epsilon$ often leads to better generalization performance. Thanks to its robustness and the empirical evidence supporting the model, we anticipate strong performance in practical applications.

## F    PROOF OF THEOREM 1

**Theorem 3.** *Suppose that $\ell(\cdot)$ is bounded by B. For some $\epsilon > 0$, if $\sum_{j=1}^{m} | \mathbb{I}(f_k(\mathbf{x}_j)) - \mathbb{I}\left(d_{\mathbf{x}_j}^{y_k}\right) |$* */m \le \epsilon$ for any $k \in [q]$ for any $\delta > 0$, with probability at least $1 - \delta$, we have*

$$R(\hat{f}) - R(f^*) \le 2qB\epsilon + 4qL_E R_N(\mathcal{F}) + 2qB\sqrt{\frac{\log\frac{2}{\delta}}{2N}}.$$

**Proof.** Before proving the theorem, we first provide two useful lemmas as follows. We primarily derive the uniform deviation bound between $R(\hat{f})$ and $R(f)$.

**Lemma 1.** *Suppose that the loss function $\ell$ is $L_E$-Lipschitz continuous with respect to $\theta$. For any $\delta > 0$, with probability at least $1 - \delta$, we have*

$$|R(\hat{f}) - \hat{R}(f)| \le 2qL_E R_{n+m}(\mathcal{F}) + qB\sqrt{\frac{\log\frac{2}{\delta}}{2(n+m)}} \tag{6}$$

*Proof.* In order to prove this lemma, we define the Rademacher complexity of $L$ and $\mathcal{F}$ with $m + n$ training examples as follows:

$$R_{n+m}(L \circ \mathcal{F}) = \mathbb{E}_{\mathbf{x},\mathbf{d},\sigma}\left[\sup_{f \in \mathcal{F}} \sum_{i=1}^{n} \sigma_i \ell\left(f(\mathbf{x}_i), \mathbf{d}_i\right) + \sum_{j=1}^{m} \sigma_j \ell\left(f(\mathbf{x}_j), \mathbf{d}_j\right)\right]$$

where $\sigma_i$ and $\sigma_j$ are Rademacher variables.

Considering that $C(f(\mathbf{x}), \mathbf{d}) = \sum_{i=1}^{m} \ell(f_k, \mathbf{d}_k)$, we have

$$R_{n+m}(L \circ \mathcal{F}) \leq q R_{n+m}(\ell \circ \mathcal{F}) \leq q L_E R_{n+m}(\mathcal{F})$$

where the second line is due to the Lipschitz continuity of the loss function $\ell$.

Then, we proceed the proof by showing that one direction $\sup_{f \in \mathcal{F}} R(f) - R(\hat{f})$ is bounded with probability at least $1 - \delta/2$, and the other direction can be proved similarly. According to *McDiarmid's inequality* (Combes, 2015), for any $\delta > 0$, with probability at least $1 - \delta/2$, we have

$$\sup_{f \in \mathcal{F}} R(\hat{f}) - R(f) \leq \sup_{f \in \mathcal{F}} R(\hat{f}) - R(f) + qB\sqrt{\frac{\log \frac{2}{\delta}}{2(n+m)}}$$

According to the result in (Mohri et al., 2018) (Theorem 3.3) that shows $\mathbb{E} \sup_{f \in \mathcal{F}} R(\hat{f}) - R(f)$ $\leq 2R_m(\mathcal{F})$, by further considering the other direction $\sup_{f \in \mathcal{F}} R(f) - R(\hat{f})$, with probability at least $1 - \delta$, we have

$$\sup_{f \in \mathcal{F}} \mid R(\hat{f}) - R(f) \mid \leq 2qL_E R_m(\mathcal{F}) + qB\sqrt{\frac{\log \frac{2}{\delta}}{2n+m}}$$

which completes the proof. □

Then, we can bound the difference between $R(\hat{f})$ and $R(f)$ as follows:

**Lemma 2.** *Suppose that $\ell(\cdot)$ is bounded by $B$. For some $\epsilon > 0$, if $\sum_{j=1}^{m} \mid \mathbb{I}(f_k(\mathbf{x}_j)) - \mathbb{I}\left(d_{\mathbf{x}_j}^{y_k}\right) \mid$ $/m \leq \epsilon$ for any $k \in [q]$ for any $\delta > 0$, we have:*

$$\mid \hat{R}_U(f) - R_U(f) \mid \leq qB\epsilon$$

*Proof.* Without loss of generality, assume that $\epsilon$ is the largest pseudo-labeling error among $q$ classes, i.e., $\epsilon = \max_{k=1}^{q} \sum_{j=1}^{m} \mid \mathbb{I}(f_k(\mathbf{x}_j)) - \mathbb{I}\left(d_{\mathbf{x}_j}^{y_k}\right) \mid /m \leq \epsilon$ for any $k \in [q]$. Obviously, $\epsilon$ consists below pseudo-labeling error:

$$\epsilon = \frac{\sum_{j=1}^{m} \mathbb{I}\left(f_k(\mathbf{x}_j), d_{\mathbf{x}_j}^{y_k}\right)}{m} \tag{7}$$

Then, we prove the following side, which provide the bounds for $R_U(f)$. Firstly, we prove its upper bound:

$$\begin{aligned}
\widehat{R}'_u(f) &= \frac{1}{m} \sum_{j=1}^{m} \sum_{k=1}^{q} \mathbb{I}(f_k(\mathbf{x}_j)) \ell(f_k(\mathbf{x}_j)) \\
&\leq \frac{1}{m} \sum_{j=1}^{m} \sum_{k=1}^{q} \mathbb{I}\left(d_{\mathbf{x}_j}^{y_k}\right) \ell(f_k(\mathbf{x}_j)) + \mathbb{I}(d_{\mathbf{x}_j}^{y_k}, f_k(\mathbf{x}_j)) \ell(f_k(\mathbf{x}_j)) \\
&\leq \frac{1}{m} \sum_{j=1}^{m} \mathcal{L}\left(f(\mathbf{x}_j), d_{\mathbf{x}_j}^{y_k}\right) + \epsilon \sum_{k=1}^{q} \ell(f_k(\mathbf{x}_j)) \\
&\leq \widehat{R}_u(f) + qB\epsilon
\end{aligned} \tag{8}$$

where the second line holds based on Eq.(7). Then, we prove its low bound:

$$
\begin{aligned}
\widehat{R}'_u(f) &= \frac{1}{m} \sum_{j=1}^{m} \sum_{k=1}^{q} \mathbb{I}\left(f_k\left(\mathbf{x}_j\right)\right) \ell\left(f_k\left(\mathbf{x}_j\right)\right) \\
&\geq \frac{1}{m} \sum_{j=1}^{m} \sum_{k=1}^{q} \mathbb{I}\left(d_{\mathbf{x}_j}^{y_k}\right) \ell\left(f_k\left(\mathbf{x}_j\right)\right) - \mathbb{I}(d_{\mathbf{x}_j}^{y_k}, f_k\left(\mathbf{x}_j\right)\ \ell\left(f_k\left(\mathbf{x}_j\right)\right) \\
&\geq \frac{1}{m} \sum_{j=1}^{m} \mathcal{L}\left(f\left(\mathbf{x}_j\right), d_{\mathbf{x}_j}^{y_k}\right) + \epsilon \sum_{k=1}^{q} \ell\left(f_k\left(\mathbf{x}_j\right)\right) \\
&\geq \widehat{R}_u(f) + qB\epsilon
\end{aligned}
\tag{9}
$$

By combining these two sides, we can obtain the following result:

$$
|\hat{R}_U(f) - R_U(f)| \leq qB\epsilon
$$

which concludes the proof.

For any $\delta > 0$, with probability at least $1 - \delta$, we have:

$$
\begin{aligned}
R(f) &\leq \hat{R}(f) + R_U(f) + 2qL_E R_{n+m}(\mathcal{F}) + qB\sqrt{\frac{\log\frac{2}{\delta}}{2N}} \\
&\leq \hat{R}(f) + R_U(f) + qB\epsilon + 2qL_E R_{n+m}(\mathcal{F}) + qB\sqrt{\frac{\log\frac{2}{\delta}}{2N}} \\
&\leq \hat{R}(f) + R_U(f) + 2qB\epsilon + 2qL_E R_{n+m}(\mathcal{F}) + qB\sqrt{\frac{\log\frac{2}{\delta}}{2N}} \\
&\leq \hat{R}(f) + R_U(f) + 2qB\epsilon + 4qL_E R_{n+m}(\mathcal{F}) + 2qB\sqrt{\frac{\log\frac{2}{\delta}}{2N}} \\
&\leq R(f) + 2qB\epsilon + 4qL_E R_{n+m}(\mathcal{F}) + 2qB\sqrt{\frac{\log\frac{2}{\delta}}{2N}}
\end{aligned}
$$

where the first and fifth lines are based on Eq. 6, and second and fourth lines are due to Lemma 1. The third line is by the definition of $f$. Putting all these together, the proof is then finished. □