# OpenReview forum: "RankMatch: A Novel Approach to Semi-Supervised Label Distribution Learning Leveraging Inter-label Correlations"
_ICLR.cc/2025/Conference — ICLR 2025 Conference Withdrawn Submission_

### Official Review · Reviewer_ov9y · 2024-11-03

**Soundness:** 2
**Presentation:** 2
**Contribution:** 2
**Rating:** 3
**Confidence:** 4

**Summary:**

This paper studies the problem of Semi-Supervised Label Distribution Learning (SSLDL) where only a proportion of samples are supervised with ground-truth label distributions, addressing the issue of high annotation cost for Label Distribution Learning (LDL). Specifically, to fully leverage the unlabeled data, the proposed method generates pseudo-label distribution using the averaged prediction from various augmented images to train the model on unlabeled data. Moreover, the proposed method adopts a pairwise relevance ranking loss to capture the correlation between labels, with a strong/weak constraint for the labeled/unlabeled samples. A generalization bound is also derived to ensure convergence for the proposed method. Experiments show the effectiveness of the proposed method.

**Strengths:**

- This paper addresses the crucial issue of high annotation cost in LDL and the proposed method outperforms the compared baselines.
- This paper provides a theoretical analysis that guarantees the convergence of the proposed method.

**Weaknesses:**

- The proposed method lacks novelty. Averaging predictions of multiple views for pseudo-labeling and pairwise ranking loss are widely adopted in existing semi-supervised learning [1] and multi-label learning methods [2, 3]. Authors are encouraged to highlight how the proposed method differs from or improves upon these prior works.
- The authors need to compare recent SOTA SSLDL or LDL methods to show the effectiveness of the proposed method (such as [4]) since the most recent baseline compared in this paper is from 2021. Or, if more recent methods were intentionally excluded, please justify your choice of baselines.
- Typos:
  - In line 028, Label Distribution Learning (LDL) (Geng, 2016) is a machine learning paradigm ...
  - In equation (4), does $q$ refer to $c$?
  - In figure 2(a)(b), Pairwise Relevance Ranking Loss
  - In line 280, Theorem 1 indicates ...
- The layout of Figure 2 and the tables is unattractive and not reader-friendly.

[1] Berthelot D, Carlini N, Goodfellow I, et al. Mixmatch: A holistic approach to semi-supervised learning[J]. Advances in neural information processing systems, 2019, 32.

[2] Gong Y, Jia Y, Leung T, et al. Deep convolutional ranking for multilabel image annotation[J]. arXiv preprint arXiv:1312.4894, 2013.

[3] Xie M K, Huang S J. Multi-label learning with pairwise relevance ordering[J]. Advances in Neural Information Processing Systems, 2021, 34: 23545-23556.

[4] Le N, Nguyen K, Tran Q, et al. Uncertainty-aware label distribution learning for facial expression recognition[C]//Proceedings of the IEEE/CVF winter conference on applications of computer vision. 2023: 6088-6097.

**Questions:**

- What does the "dual-phase training" mean in line 242?
- In the ablation of parameter $\lambda$, why 0.005\~0.1 are chosen to be analyzed but not 0\~1?

---

### Official Review · Reviewer_SzqQ · 2024-11-03

**Soundness:** 2
**Presentation:** 2
**Contribution:** 2
**Rating:** 3
**Confidence:** 3

**Summary:**

This paper proposes a semi-supervised label distribution learning method (SSLDL), RankMatch. RankMatch incorporates a novel relevance ranking loss to manage label correlations effectively.  A theoretical generalization bound is established for RankMatch.

**Strengths:**

This paper is well-structured and easy to follow. The analysis in the experimental section is relatively extensive.

**Weaknesses:**

1. The proposed method lacks of novelty, and the performance seems limited.
2. The compared methods are outdated. Please include more recent methods to prove the effectiveness.
3. Figures 1(a) and 1(b) seem to be inconsistent.
4. The format of the Appendix section is disorganized, and the corresponding citations in the main paper are incorrect.
5. The caption and content of Table 1 do not match.

**Questions:**

See weakness.

---

### Official Review · Reviewer_GdA8 · 2024-11-04

**Soundness:** 3
**Presentation:** 2
**Contribution:** 2
**Rating:** 5
**Confidence:** 4

**Summary:**

This manuscript proposes RankMatch to cope with the semi-supervised label distribution (SSLDL) problem. Specifically, RankMatch ﻿incorporates a advanced PRR loss for LDL problem to capture the inter-label correlations. This manuscript is overall good and well-written, and I have the following concerns and suggestions:

**Strengths:**

This manuscript is overall well-written

**Weaknesses:**

1)One flowchart is encouraged for better comprehension.
2)In page 4, as the manuscript stated, if the description degree of two labels are close, they are regarded as positively correlated. Is this means the correlated from the semantic perspective? For example, in the natural scene image, the similar values only represent the similar description proportion in this image. It is encouraged to be further explained.
3)According to [1], utilizing strong augmentations, like Cutout, may lead the distribution-shifting for the input image. Authors are encouraged to discuss whether this strategy affect the self-training of unlabeled data.
4)RankMatch introduces the margin to penalize the label ranking. Actually, from the perspective of interpretation, LDL framework is sometimes appropriate for the tasks with large label dimension equipped with complex semantic, of which the degree differences among candidate labels may be small. Like the data set Yeast-alpha, the degree differences are typically smaller than 0.01 across instances. Authors are encouraged to discuss this situation.
5)Authors are encouraged to conduct a simple comparison experiment on MLL data sets, like cifar-10.
6)In the APPENDIX, the part of Implementation is repeated.
[1]. Tack J, Mo S, Jeong J, et al. Csi: Novelty detection via contrastive learning on distributionally shifted instances[J]. Advances in neural information processing systems, 2020, 33: 11839-11852.

**Questions:**

1)One flowchart is encouraged for better comprehension.
2)In page 4, as the manuscript stated, if the description degree of two labels are close, they are regarded as positively correlated. Is this means the correlated from the semantic perspective? For example, in the natural scene image, the similar values only represent the similar description proportion in this image. It is encouraged to be further explained.
3)According to [1], utilizing strong augmentations, like Cutout, may lead the distribution-shifting for the input image. Authors are encouraged to discuss whether this strategy affect the self-training of unlabeled data.
4)RankMatch introduces the margin to penalize the label ranking. Actually, from the perspective of interpretation, LDL framework is sometimes appropriate for the tasks with large label dimension equipped with complex semantic, of which the degree differences among candidate labels may be small. Like the data set Yeast-alpha, the degree differences are typically smaller than 0.01 across instances. Authors are encouraged to discuss this situation.
5)Authors are encouraged to conduct a simple comparison experiment on MLL data sets, like cifar-10.
6)In the APPENDIX, the part of Implementation is repeated.

[1]. Tack J, Mo S, Jeong J, et al. Csi: Novelty detection via contrastive learning on distributionally shifted instances[J]. Advances in neural information processing systems, 2020, 33: 11839-11852.

---

### Official Review · Reviewer_LDiy · 2024-11-11

**Soundness:** 3
**Presentation:** 3
**Contribution:** 3
**Rating:** 5
**Confidence:** 4

**Summary:**

The paper proposes a pairwise relevance ranking loss which captures inter-label correlations to tackle Semi-supervised Label Distribution Learning.

**Strengths:**

The proposed framework employs an ensemble of loss function to handle semi-supervised label distribution learning. The introduction of pairwise relevance looks simple and effective in the reported results.
Overall, the paper is easy to read.

**Weaknesses:**

1.	While the proposed method introduces an ensemble of loss functions such as a supervised loss term as KL divergence, an unsupervised consistency loss as considering probability distribution for weakly augmented version of unlabeled image and a pairwise relevance ranking loss for optimization; the paper does not provide computational costs for the framework. It is also absolutely crucial to find out how the proposed framework will work in case of using a complex , larger deep model architecture.

2.	It’d be very interesting to see how the framework would work with large set unlabeled samples using some well-known large-scale dataset such as ImageNet.

**Questions:**

Please refer to the weaknesses section.

---

### Note · Authors · 2024-11-12

I have read and agree with the venue's withdrawal policy on behalf of myself and my co-authors.